REGISTERED REPORT PROTOCOL

# Availability, acceptability and uptake of Sexual Reproductive Health interventions for young people with disabilities in Sub Saharan Africa: A scoping review protocol

Itai Kabonga[ID]*, Tapson Mashanyare*, Owen Nyamwanza[ID]

Centre for Sexual Health and HIV/AIDS Research- Zimbabwe, Belgravia, Harare, Zimbabwe

* itai.kabonga@ceshhar.org

## Abstract

### Introduction

Young people with disabilities face major barriers in accessing sexual reproductive health (SRH) services in resource-poor settings, including Sub Saharan Africa (SSA). Although, there is increasing recognition of their unique SRH needs, the availability, acceptability, and uptake of SRH service delivery interventions for this population group remain understudied. Young people with disabilities encounter barriers to accessing SRH services due to stigma, poverty, lack of information and physical barriers. We aim to map existing literature on SRH service delivery interventions targeting young people with disabilities in SSA through a scoping review.

### Methods and analysis

The scoping review will be guided by the Arksey and O'Malley methodological framework. Articles will be searched in PubMed, African Index Medicus, Google Scholar, African Journals Online, Web of Science and Embase electronic databases as well as grey literature database, Open Grey. We will also do a citation search of references of eligible papers for literature that may have been overlooked in other searches. A two-step process will be used to screen retrieved articles i) title and abstract screening ii) full text screening. Results of the scoping review will be reported following the Preferred Reporting Items for Systematic Reviews and Meta-Analyses (PRISMA-P) extension for scoping reviews.

### Discussion

There is a paucity of knowledge on availability, acceptability and uptake of SRH service delivery interventions for young people with disabilities in SSA. This scoping review is poised to fill the gap by demonstrating the breadth of literature on availability, acceptability and uptake of SRH service delivery interventions for young people with

This is a Registered Report and may have an associated publication; please check the article page on the journal site for any related articles.

**Data availability statement:** No data is associated with the current submission.

**Funding:** The author(s) received no specific funding for this work.

**Competing interests:** The authors have no competing interests.

disabilities The scoping review aims to map the availability of SRH service delivery interventions for young people with disabilities. This mapping of evidence has the potential to identify whether there is a need for SRH service delivery interventions for young people with disabilities. Our scoping review will map which SRH service delivery interventions work or do not work, as well as gaps in SRH service delivery interventions for young people with disabilities. This information is useful for policy making and for designing effective SRH service delivery interventions for young people with disabilities.

## Introduction

The World Health Organisation (WHO) estimates there are 1.3 billion people living with disabilities globally in 2025, including 220 million young people [1,2]. Majority of these young people, over 80%, live in low-income countries including Sub-Saharan Africa (SSA) [3]. Young people with disabilities (YPWD) have the worst socio-economic and health outcomes [4–6]. They are most likely to be living in poverty, attained low levels of education, living in poor environments and/or unemployed [4,5,7,8]. YPWD are much less likely to be screened for HIV, sexually transmitted infections (STIs), cervical and breast cancer, less likely to have received comprehensive sexuality education and less likely to have access to contraception [9,10]. The lack of information and scarcity of accessible services exacerbates their vulnerability [11].

While the provision of SRH services to people with disabilities is a constitutional right in many SSA countries and an international obligation under United Nations Convention on the Rights of Persons with Disabilities (CRPD) and International Covenant on Economic, Social and Cultural Rights (ICESR), access to services remains a challenge [4,8,12]. Although young people in general have unmet SRH needs [4,13], YPWD are worse off as a result of multiple intersecting barriers which prevent access to SRH services.

Evidence from SSA demonstrates several SRH service delivery interventions for young people in general [13]. SRH service delivery interventions for young people in general in SSA focus on HIV prevention and treatment, maternal services, sexual health education, STI services, contraception services, peer education on SRH issues and many others [12–16]. There is also evidence of availability of SRH service delivery interventions for YWPD focusing on provision of sexuality education, HIV prevention and treatment and support for YWPD who survive sexual and domestic violence [17,18]. Kassa et al., show few national programs in Ethiopia seeking to address the SRH needs of YPWD [19]. In most countries in SSA, Non-governmental organisations (NGOs) are usually front runners in implementing SRH service delivery interventions targeting YPWD [17,18]. However, NGO-driven interventions tend to have low coverage and are often not sustainable [20].

Extant literature identifies several barriers which limit YPWD from accessing SRH services. YWPD are stigmatised and considered asexual; a misconception widely shared among health workers and the general population alike [4,21–23]. YPWD fail to access SRH services because of inaccessible health facilities, insensitivity of health

care workers, limited information on disability among health care workers, and lack of information by YPWD [4,11,24–27]. YPWD are further excluded from accessing SRH services due to transport challenges, user fees, long waiting times, need for an assistant and lack of confidentiality [28,29]. YPWD are perceived by some health workers as an additional burden to an already burdened health system [30–32]. This emanates from preconceived notions on disabilities, causing some health workers to view YPWD as more challenging for care, making them less keen to offer services [19,33].

The way in which majority of SRH service delivery interventions are designed potentially limits access to SRH services by YPWD [19]. It is assumed that SRH service delivery interventions designed for general young people, will somehow reach YPWD [19]. This belief fails to appreciate the unique experiences of YPWD, and this has motivated some non-state actors such as NGOs to design SRH interventions for YPWD. Care should be taken when designing SRH service delivery interventions for young people and ensure they are inclusive of YPWD [34]. SRH service delivery interventions for YWPD potentially have better outcomes when they are included in the design, implementation, monitoring and evaluation [1,22].

Demonstrated above are several challenges faced by YPWD in accessing SRH services. We have designed this scoping review to explore the breadth of literature on availability, acceptability and uptake of SRH service delivery interventions for YPWD. It transcends other studies which focused on people with disabilities in general [10], women with disabilities [35] and specific disabilities [8]. The scoping review will generate evidence on SRH programming for YPWD, mapping evidence on interventions that work well or do not work well and gaps in SRH service provision. This evidence is useful to practitioners and policy makers in addressing unmet SRH needs for YPWD.

## Methods

### Protocol design

Our scoping review will employ the framework developed by Arksey and O'Malley [36] consisting of five stages i) identifying the research questions, ii) identifying the relevant studies, iii) study selection, iv) charting the data, iv) collating, summarizing and reporting the results. The presentation of the findings will be informed by the PRISMA-P. We registered the protocol with Open Science Framework, on the 17th of December 2024, https://osf.io/6mubr.

**Stage 1: Identifying the research questions.** The research questions were developed through team consultation, and the team concurred that the research questions were adequate to explore the range of literature on availability, acceptability and uptake of SRH service delivery interventions for YPWD in SSA. The specific research questions for the scoping review are:

• What are the available SRH service delivery interventions for YPWD in SSA?

• How acceptable are SRH service delivery interventions for YPWD in SSA?

• What has been the uptake of SRH service delivery interventions for YPWD in SSA?

The research questions were developed guided by the population-concept and context mnemonic, Table 1:

**Table 1. Outline of population, concept and context.**

| Framework category | Items |
|---|---|
| Population | • Young people with disabilities (10–24 years) |
| Concept | • HIV prevention and treatment<br>• Contraception services<br>• Maternal services<br>• Abortion services<br>• STIs services<br>• Menstrual services<br>• Sexual health education |
| Context | • Sub Saharan Africa<br>• People living with disability |

**Stage 2: Identifying relevant studies.** We are going to identify relevant studies through i) searching of electronic databases and grey literature databases ii) searching of identified studies' bibliographies for relevant studies. We will search PubMed, African Index Medicus, Google Scholar, African Journals Online, Embase and Web of Science electronic databases for relevant studies. Additionally, we are going to search Open Grey database for reports, doctoral dissertations, official publications and other types of grey literature. The databases will be searched using search terms combined with operator 'OR', Table 2. Before the comprehensive search, we will pretest the search terms in PubMed to gauge sensitivity.

**Stage 3: Study selection.** The search results from stage 2 will be exported to Endnote for data management, specifically for removal of duplicates and referencing. The deduplicated bibliography will be exported to Covidence for title and abstract screening. Title and abstract screening will be conducted by two independent reviewers (IK and TM) using the inclusion and exclusion criteria below, Table 3. The disagreements between the two reviewers will be resolved through reviewers' discussion or arbitration by a third person (ON). Selected studies will be subjected to full text screening using the inclusion and exclusion criteria. Full text screening will be conducted by two independent reviewers (IK and TM) and disagreements will be settled by reviewers through discussion or arbitration by third person (ON) when unresolved. Studies meeting the inclusion criteria will be retrieved and those not meeting the inclusion criteria will be excluded.

**Stage 4: Charting the data.** We will develop an Excel data charting tool to capture study characteristics (author, year of publication, country of publication, study design, methodology of developing SRH interventions) and key issues and themes related to scoping review objectives (for instance acceptability of SRH service delivery interventions, uptake of SRH service delivery interventions, gaps in SRH service delivery interventions, and other studies key findings). Initially, we will pilot the charting tool on 10 randomly selected studies and assess its usability and comprehensiveness. The charting process will be iterative, and the tool will be modified as necessary. During the charting process, we may contact the authors of studies for clarification of study characteristics.

**Stage 5: Collating, summarizing, and reporting the results.** The PRISMA flow diagram will visualize the included and excluded studies. Synthesis of data will follow the guidance provided by Levac et al., [37]. We will analyse data on study characteristics and summarize it using descriptive statistics. Additionally, we will deploy thematic analysis approach and summarize narratively the data relating to availability, and acceptability of SRH service delivery interventions for YPWD. Reporting will be iterative, with continuous adjustment of reporting. We will also discuss the meanings of our findings including implications for policy, practise and future research

## Ethics and dissemination

The study will not seek ethical approval as we will be using studies and grey literature that has already been published. We will disseminate the findings of the study through publication of the scoping review in a peer-reviewed journal as well as presenting at relevant conferences. The findings will be shared with other researchers in our network and stakeholders in the SRH field.

## Discussion

Several studies highlight SRH service delivery interventions targeting young people in general [15,38,39]. However, there is a paucity of knowledge on availability, acceptability and uptake of SRH service delivery interventions for YPWD. Our scoping review is posed to fill the gap by demonstrating the breadth of literature on availability, acceptability and uptake of SRH service delivery interventions for YPWD. Currently it remains unknown which SRH service delivery interventions for YPWD are available. Mapping evidence of availability of interventions is needed to make recommendations to policy makers and programmers whether there is any need of specific SRH service delivery interventions for YPWD. Additionally, our scoping review will map which interventions work and do not work as well as gaps in SRH service delivery interventions for YPWD. This consolidated evidence will be useful to inform policymakers and practitioners in designing effective SRH

**Table 2. Search strategy.**

| Concept | Search Terms |
|---|---|
| Disability | Disabled OR "Disabled Person*" OR "Living with disab*" OR "Persons with disab*" OR "People with disabilit*" OR "Persons with hearing disab*" OR "Persons with intellectual disabilit*" OR "Persons with visual disab*" OR "Intellectual disab*" OR "Communication disorder*" OR "Developmental disab*" OR "Mentally Disabled person*" OR "Mental Disab*" OR "functional impairment*" OR "mobility impairment" OR "sensory impairment" |
| Sexual and reproductive health | "Sexual behavi*" OR "sexual health" OR "Youth sexual behavi*" OR "Attitudes toward sex" OR Sexualit* OR "Sexual Health -- In Young People" OR "reproductive health" OR "reproductive health service* "Reproductive Health -- In Young People" OR contracepti* OR "family planning" OR "family planning service*" OR "Protected sex" OR HIV OR STI* OR "sexual transmitted*" OR "Abortion service*" OR "Menstrual health" OR "Sexual Health Education" |
| Young People | Teen* OR "Young adult*" OR "Emerging adult*" OR Youth* OR "young person" OR "10-24 years old" OR "young adolescent*" OR "Very young adolescent*" OR "early adolesc**" OR adolescence OR adolesc* OR teenag* OR "Female adolescent*" OR "female adolesc*" OR "male adolesc*" |
| Sub-Saharan Africa | Angola OR Benin OR Botswana OR "Burkina Faso" OR Burundi OR Cameroon OR "Cape Verde" OR "Central African Republic" OR CHAD OR Comoros OR Congo OR "Congo Democratic Republic" OR Djibouti OR "Equatorial Guinea" OR Eritrea OR Ethiopia OR Gabon OR Gambia OR Ghana OR Guinea OR "Guinea-Bissau" OR "Cote d'Ivoire" OR "Ivory Coast" OR Kenya OR Lesotho OR Liberia OR Madagascar OR Malawi OR Mali OR Mozambique OR Namibia OR Niger OR Nigeria OR "Sao Tome and Principe" OR Rwanda OR Senegal OR Seychelles OR "Sierra Leone" OR Somalia OR "South Africa" OR "South Sudan" OR "Sudan" OR "Swaziland" OR "Eswatini" OR "Mauritania" OR "Mauritius" OR "Tanzania" OR "Togo" OR "Uganda" OR "Zambia" OR "Zimbabwe" OR "West Africa" OR "East Africa" OR " Central Africa" OR " Southern Africa" OR "Africa South of the Sahara" OR "sub-Saharan Africa'' |

Concepts search terms will be combined by operator "AND".

**Table 3. Inclusion and Exclusion criteria.**

| | Inclusion | Exclusion |
|---|---|---|
| Focus | • HIV prevention and treatment interventions for YPWD<br>• Contraception interventions for YPWD<br>• Maternal health interventions for YPWD<br>• Abortion interventions for YPWD<br>• STIs interventions for YPWD<br>• Comprehensive sexuality education for YPWD<br>• Menstrual health interventions for YPWD<br>• Sexual health education for YPWD | • SRH service delivery interventions for young people without disabilities |
| Design | Qualitative, Quantitative, Mixed methods | Literature reviews (scoping, systematic, meta-analysis etc) |
| Language | English | Not in English |
| Context | Sub Saharan Africa | Europe, North America, South America, Asia, Middle East, Oceania |
| Timeline | January 2000 to April 2025 | Before 2000 |

service delivery interventions to reach YPWD. Based on the findings of the scoping review, we will also be able to recommend any areas needing primary research.

Potential limitation of the study is exclusion of studies not in English as we are constrained in terms of resources to include studies in other languages. We are aware that exclusion of non-English papers potentially introduce bias. Though there is option to use Google Translate for studies not in English, we worry that it may not always accurately convey the nuances of scientific terminology or complex ideas, which may potentially lead to misinterpretations of concepts. However, we believe identified English studies in SSA will be a useful gauge of the breadth of literature on availability, acceptability and uptake of SRH service delivery interventions for YPWD.

## Conclusion

Our study will fill knowledge gaps on availability, acceptability and uptake of SRH service delivery interventions for YPWD. The scoping review has potential to generate insights that can be utilized in developing SRH service delivery interventions for YPWD.

## Supporting information

**S1. PRISMA-P checklist.**
(DOCX)

## Author contributions

**Conceptualization:** Itai Kabonga, Tapson Mashanyare*.

**Supervision:** Owen Nyamwanza.

**Writing – original draft:** Itai Kabonga, Tapson Mashanyare*, Owen Nyamwanza.

**Writing – review & editing:** Itai Kabonga, Tapson Mashanyare*, Owen Nyamwanza.

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
