## [Decision Letter · Decision Letter 0]

5 Mar 2025

PONE-D-25-02097Availability, acceptability and uptake of Sexual Reproductive Health interventions for young people with disabilities in Sub Saharan African: A scoping review protocol

PLOS ONE

Dear Dr. kabonga,

Thank you for submitting your manuscript to PLOS ONE. After careful consideration, we feel that it has merit but does not fully meet PLOS ONE’s publication criteria as it currently stands. Therefore, we invite you to submit a revised version of the manuscript that addresses the points raised during the review process.

We look forward to receiving your revised manuscript.

Kind regards,

Ayobami Precious Adekola, PhD in Public Health

Academic Editor

PLOS ONE

2. In your cover letter, please confirm that the research you have described in your manuscript, including participant recruitment, data collection, modification, or processing, has not started and will not start until after your paper has been accepted to the journal (assuming data need to be collected or participants recruited specifically for your study). In order to proceed with your submission, you must provide confirmation.

Additional Editor Comments:

Dear Dr Kabonga,

We have received the reports from our reviewers on your manuscript. Based on the advice received, your manuscript can be accepted for publication after you have carried out the corrections as suggested by the reviewers. I look forward to receiving the revised manuscript based on the feedback.

Best regards,

Dr Adekola

Reviewers' comments:

Reviewer's Responses to Questions

**Comments to the Author**

1. Does the manuscript provide a valid rationale for the proposed study, with clearly identified and justified research questions?

Reviewer #1: Yes

Reviewer #2: Yes

Reviewer #3: Yes

2. Is the protocol technically sound and planned in a manner that will lead to a meaningful outcome and allow testing the stated hypotheses?

Reviewer #1: Yes

Reviewer #2: Yes

Reviewer #3: Yes

3. Is the methodology feasible and described in sufficient detail to allow the work to be replicable?

Reviewer #1: Yes

Reviewer #2: Yes

Reviewer #3: Yes

4. Have the authors described where all data underlying the findings will be made available when the study is complete?

Reviewer #1: Yes

Reviewer #2: Yes

Reviewer #3: Yes

5. Is the manuscript presented in an intelligible fashion and written in standard English?

Reviewer #1: Yes

Reviewer #2: Yes

Reviewer #3: Yes

6. Review Comments to the Author

You may also provide optional suggestions and comments to authors that they might find helpful in planning their study.

Reviewer #1: Review of the Article

PONE-D-25-02097

Availability, acceptability and uptake of Sexual Reproductive Health interventions for young people with disabilities in Sub Saharan African: A scoping review protocol

PLOS ONE

I want to congratulate the authors for selecting an interesting topic. The manuscript is well-written. However, I would like to provide a few suggestions for revising the manuscript.

Title and Abstract

The title accurately reflects the focus of the study: “Availability, acceptability, and uptake of sexual reproductive health interventions for young people with disabilities in Sub-Saharan Africa.” However, there is a minor grammatical issue in the title: "African" should be "Africa" (i.e., Sub-Saharan Africa instead of Sub-Saharan African).

The abstract clearly states the research problem, methodology, and expected contribution of the study. However, the research gap could be articulated more strongly. For instance, instead of broadly stating that knowledge is limited, the authors should specify which aspects of SRH interventions remain particularly underexplored (e.g., policy gaps, service delivery models).

The discussion in the abstract could better explain the practical implications of the study, particularly how it informs policy and programming.

Introduction

The introduction effectively frames the importance of SRH services for young people with disabilities. It provides statistical data and international policy references (e.g., CRPD, ICESCR), making it a strong foundation. However, some areas need improvement:

i. Several points (e.g., stigma, inaccessibility of SRH services, exclusion from interventions) are mentioned multiple times. A more concise presentation of these barriers would improve readability.

ii. Concepts like "buy-in," "existential realities," and "coterie of challenges" could be simplified or replaced with more precise language to enhance accessibility.

iii. The introduction focuses on challenges or barriers to accessing SRH interventions (stated broadly) without addressing availability as an important variable. The study fails to highlight the existing interventions for the general population and those specific to young people with disabilities. Discussing availability first would provide a clearer framework for assessing accessibility and mapping out which interventions work and which do not. The scoping review would be strengthened by incorporating this important variable.

Methodology

The methodology is robust and follows the Arksey and O’Malley scoping review framework. However, a few areas need clarification:

While a list of databases is provided, the authors should include details on:

i. How search terms were combined using Boolean operators (e.g., AND, OR).

ii. Whether any language restrictions were applied beyond English.

iii. Whether gray literature sources (e.g., NGO reports, government documents) will be systematically appraised.

The authors state that two reviewers will independently screen articles, but how will discrepancies be handled if the third reviewer is unavailable?

Will any software (e.g., Rayyan, Covidence) be used to ensure consistency in the review process?

The authors mention CASP checklists and RoB tools, but scoping reviews typically do not conduct quality appraisal. They should clarify why these tools are relevant in this context.

The manuscript states that no ethical approval is required because only secondary data is used. However, it would be beneficial to acknowledge any potential ethical concerns regarding the representation of people with disabilities in research.

Discussion

The discussion explains the gaps in SRH services for young people with disabilities and the potential contribution of the review. However:

The statement: "Our study will fill knowledge gaps on availability, acceptability, and uptake of SRH interventions for young people with disabilities." is overly broad. This gap has not been clearly established based on the review, and it needs to be explicitly defined. The authors should specify which aspects of these interventions they aim to address, as previously mentioned. Importantly, a scoping review does not evaluate effectiveness; it maps available literature.

The exclusion of non-English studies is noted as a limitation, but the authors should discuss how this might bias the findings (e.g., missing key studies from Francophone and Lusophone African countries). This exclusion could lead to an incomplete review.

"There is paucity of knowledge" should be corrected to "There is a paucity of knowledge."

Some long sentences could be broken down for better clarity. The paper would benefit from an editorial review for grammatical errors and sentence structure improvements.

Some claims, especially regarding barriers to SRH services, stigma, and discrimination, lack proper citations. Every statement making a claim about accessibility or service delivery should be backed by references.

The manuscript does not explicitly outline areas for further research. A strong scoping review should conclude with recommendations on which SRH interventions need further study, the need for primary research in specific areas where literature is lacking, and how policies and programmes should be tailored based on the findings.

The manuscript is well-structured and covers an important topic. However, addressing the issues of availability, clarifying methodological details, tightening the discussion, and ensuring proper citation of claims will significantly improve the quality of the paper. Additionally, a final editorial review for grammar and clarity is recommended.

Reviewer #2: Thank you for the opportunity to review this manuscript. Overall, this is an interesting topic and a well-written manuscript. I raised some issues and provided recommendations below:

INTRODUCTION

1. Provide a date/year for this data: “The World Health Organisation (WHO) estimates that there are 1 billion people living with disabilities globally, including 220 million young people.”

2. Paragraph 2 is too long. A paragraph may not exclude 12 lines.

3. This statement appears unclear: “SRH interventions for young people with disabilities fail because of lack of buy-in as well as being divorced from their contextual experiences and realities of living with disabilities.”

METHODS

4. The disability context should include the “people living with disability” keyword.

5. You may have to include MeSH terms for the constructs. For example, disability has the "Persons with Disabilities"[Mesh] MeSH term, while sub-Saharan Africa has a MeSH in PubMed: "Africa South of the Sahara"[Mesh].

6. There is a need to incorporate * to explode some terms. For example, some authors may use the phrase “sexual behaviour” (British language), while others may use “sexual behavior” (American English). To capture both simultaneously, you may include the * and write it as “sexual behavi*”. Another example is youth. I would write youth* to capture both youth and youths.

7. The number of countries in the sub-saharan Africa concept is incomplete. Eswatini, Mauritania, and Mauritius are missing. Furthermore, please include “West Africa,” “East Africa,” “Southern Africa,” OR “Central Africa” because some authors focus on multiple countries within the same region and report regions rather than individual countries.

8. Single terms such as disabled, Congo, etc. do not need to be in quote. “ “ is used for two or more words to make them “one” or turn them into phrases.

9. Please note that Embase uses emtree (similar to MeSH) terms. Please use the available emtree in Embase for a more comprehensive search.

10. I can see from Table 2 that the authors plan to remove non-English articles. I strongly discourage that because there are tools that can translate documents now. Google Translate does a good job, too. So, this is no longer a valid exclusion criterion because non-English articles matter.

11. Is there a justification for limiting the search to 2000-2025?

DISCUSSION

12. I discussed the issue of excluding non-English studies.

Thank you.

Reviewer #3: I appreciate the invitation to perform this review. In my opinion, the protocol was satisfactorily written with all the necessary components needed included and detailed information of step by step procedures of the methodology included. it is a well written protocol explicit and detailed.

7. PLOS authors have the option to publish the peer review history of their article (what does this mean? ). If published, this will include your full peer review and any attached files.

**Do you want your identity to be public for this peer review?** For information about this choice, including consent withdrawal, please see our Privacy Policy .

Reviewer #1: No

Reviewer #2: **Yes: ** Oluwaseun Abdulganiyu Badru

Reviewer #3: **Yes: ** Professor Esther Olufunmilayo Asekun-Olarinmoye

---

## [Editor Report · Decision Letter 1]

24 Apr 2025

Availability, acceptability and uptake of Sexual Reproductive Health interventions for young people with disabilities in Sub Saharan African: A scoping review protocol

PONE-D-25-02097R1

Dear Dr. Kabonga,

We’re pleased to inform you that your manuscript has been judged scientifically suitable for publication and will be formally accepted for publication once it meets all outstanding technical requirements.

Kind regards,

Ayobami Precious Adekola, PhD in Public Health

Academic Editor

PLOS ONE
---

## [Editor Report · Acceptance letter]

PONE-D-25-02097R1

PLOS ONE

Dear Dr. kabonga,

I'm pleased to inform you that your manuscript has been deemed suitable for publication in PLOS ONE. Congratulations! Your manuscript is now being handed over to our production team.

Kind regards,

on behalf of

Dr. Ayobami Precious Adekola

Academic Editor

PLOS ONE